# AUGMENTING COLLECTIVE INTELLIGENCE THROUGH BELBIN'S TEAM ROLES

## 1 INTRODUCTION

One of the most important inputs into an organization's performance is the efficacy and well-being of its collectives [1]. Successful collectives demonstrate not just individual but collective intelligence (CI). This emergent property allows groups to solve various problems together more effectively than individuals could on their own (Woolley et al., 2015). Studying CI can predict how fast groups learn, perform, and improve performance together. Augmented CI (ACI) combines AI systems and humans into collectives that offer tremendous potential to realize the benefits of the intellectual revolution (Foster, 2023) for organizations.

### 1.1 AUGMENTING THE BELBIN TEAM ROLES

Human collectives can be divided into near-infinite combinations, given that each collective's dynamics change based on the tasks and relationships at hand (Fisher et al., 1998). For our analysis, we will use Belbin's Team Roles (Belbin, 1993), a list of nine "clusters of behavioral attributes" that drive collective performance. The nine roles cover three dimensions: action, thought, and human relationships. We offer a novel approach to the nine roles by examining each for its potential to be augmented by AI systems, i.e., how these roles should be envisioned in an ACI context. Each role will be rated as having low, medium, or high potential for AI to augment or replace.

### 1.2 THE FUTURE OF HYBRID TEAMING

To be competitive in this decade, organizations must adopt robust strategies for ACI, including integrating AI tools into hybrid collectives alongside humans. Doing so requires matching collective compositions with task domains correctly – and understanding in detail which roles might be played well by humans or machines. AI system capabilities will continue to increase rapidly; a sound ACI strategy must capture an accurate snapshot of current dynamics and consider how capabilities are likely to evolve Gruetzemacher et al. (2021).

## 2 BELBIN'S NINE ROLES

We define and evaluate the nine roles outlined by Belbin, including the action roles (Shaper, Implementer, and Completer-Finisher), the thinking roles (Plant, Monitor-Evaluator, and Specialist), and the human relationship roles (Resource Investigator, Teamworker and Co-ordinator).

### 2.1 ACTION ROLES

**Shaper** – *Current augmentation potential: Low*
This role includes executive responsibilities such as setting the vision, overseeing collectives, and managing crises as they come up. Given the high level of abstraction and uncertainty and the broad domains this role covers, it is the least well-suited for AI augmentation and, for the time being, can be considered "irreplaceable." As AI systems improve in strategic coordination and become more agentic, possibilities for using an AI system as a Shaper increase. However, this may take time and require developers to solve complex safety problems (Hendrycks et al., 2021).

---

[1] https://www.gallup.com/workplace/215924/well-being.aspx

**Implementer –** *Current augmentation potential: High*
Implementers are analysts who make their collectives more efficient; they have high follow-through, attention to detail, and are highly reliable. For designers of hybrid collectives, augmenting the Implementer role is an obvious choice. AI system performance is superior to humans in many of these areas.

**Completer-Finisher –** *Current augmentation potential: High*
Completer-Finishers debug work and perform quality control, raising their collective's performance standards. Many AI tools already perform this function. Grammarly (Zhang et al., 2023), for example, automates error identification and adjusts tone in written work.

## 2.2 THINKING ROLES

**Plant –** *Current augmentation potential: Medium*
Plants are creative thinkers who contribute to collective brainstorms. Creativity, while long thought to be humans' exclusive domain, is an area where novel AI tools show much promise. Using generative AI (GenAI) tools to supplement inputs to collectives' ideations can increase motivation and efficiency by helping to overcome the "blank page problem" (Davenport & Mittal, 2022).

**Monitor-Evaluator –** *Current augmentation potential: Low*
Monitor-evaluators are strategic problem solvers, adept at planning and assessing ideas' feasibility. Strategic planning has long been challenging to automate, but breakthroughs in the past two years seem promising. AI systems are now outperforming humans in strategy games like Diplomacy (, FAIR) and building strong economic models (Zheng et al., 2021).

**Specialist –** *Current augmentation potential: Mixed*
Specialists are individual contributors with domain-specific technical skills, e.g., technical writing, programming, or beyond. Writers and programmers have tools like copy.ai [2] or GitHub Copilot (Ortin et al., 2023), but these may be spotty and unsuitable for all specialties.

## 2.3 RELATIONSHIP ROLES

**Coordinator –** *Current augmentation potential: Low*
At first glance, the Coordinator role seems hard to augment. This people-focused role uses communication skills to motivate and encourage collaboration within the collective. Still, project management and people management software have made significant strides in using granular data collection and gamification to promote desirable behaviors. [3]

**Teamworker –** *Current augmentation potential: Medium*
Teamworkers are mediators: strong listeners who can guide teammates safely through conflict and ensure all voices are heard. AI systems don't have the same kind of EQ as humans, but some AI tools are performing well as mediators (Höne, 2019) and therapists (Fulmer et al., 2018) and have potential as Teamworkers.

**Resource Investigator –** *Current augmentation potential: Medium*
Resource Investigators build partnerships, conduct outreach, and generally build new connections between the collective and the outside world. This role is equal parts research, relationship management, and sales; augmenting it would require integrating several AI tools to match each function. For example, recommender systems could filter for potential partners or customers (Davenport et al., 2021), smart calendars could facilitate conversations (Rebelo, 2023), and GenAI tools could create messages and assets to move the relationship forward.

---

[2] https://www.copy.ai
[3] https://www.besci.org/tactics/gamification

# 3   CONCLUSION

Belbin's Team Roles offer a helpful frame to achieve the following: evaluate the competitive advantages of humans and AI tools, uncover trends, and improve collective resource allocation. Our research showed that some roles (Shaper, Monitor-Evaluator, Coordinator) would be hard to augment, while others (Implementer, Completer Finisher) would be easier. As AI capabilities inch closer to AGI (Altman, 2023), further research is needed to determine which roles will be most challenging to augment and over what time horizon.

## URM STATEMENT

The authors acknowledge that at least one key author of this work meets the URM criteria of ICLR 2023 Tiny Papers Track.

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
