# OpenReview forum: "Augmenting Collective Intelligence through Belbin’s Team Roles"
_ICLR.cc/2023/TinyPapers — Submitted to Tiny Papers @ ICLR 2023_

### Official Review · Reviewer_mhMG · 2023-03-31

**Confidence:** 4

**Summary Of Contributions:**

This paper analyzes popular behavioral attributes found in team roles in a human-AI teaming scenario. Here, they describe the potential role of AI systems in these teams as they would fit into each role, and discuss the current potential for AI systems to be augmented into each role.

**Rating:**

High Potential (HP): a submission which meets the reviewing criteria and has potential to make an impact on the field

**Strengths And Weaknesses:**

I really enjoyed reading this paper! As human-AI teams become more prevalent, I think it is important to use and adapt frameworks to help us describe where AI fits into these teams and the bounds of the role it will play. I especially enjoyed the addition of the current augmentation potential for each role.

Strengths:
- Clarity: Ideas and conclusions are presented and communicated clearly, relevant related work is well discussed.

- Correctness: The methodology, reasoning, and related work well supports the claims in the paper.

- Reproducibility: Ideas are well supported and communicated, conclusions could be reproduced.

Weaknesses:
- Follows basic requirements: The paper is over the 2 page maximum limit

**Suggested Changes:**

The paper is unfortunately over the maximum 2 page limit (the conclusion lands on the 3rd page). However I believe that the page requirement could be met without sacrificing any of the paper content. I would suggest the following: since the footnotes are just referencing websites, change them to citations. After that, if space is still needed then the subsection headers in the introduction could be removed or changed to short section headers (as in Section 2.1-2.3).

---

### Meta-Review · Area_Chair_9DWU · 2023-04-07

**Recommendation:** Invite to revise
**Confidence:** 4

**Metareview:**

1. Clarity: Though the communication is clear, relevant literature links are provided, relevant proofs are missing
2. Correctness: Author needs to improve conclusion and provide proper justification
3. Reproducibility: Considering this paper is on theoretical findings, relevant proofs missing for the assumptions of 6 out of 9 Belbin Team Roles.
4. The 3 Proof majorly provided are of applications and discussions made in websites, it would be beneficial to provide some research paper proofs as well.


Pros: \
A good start to analyse managerial book, [Team roles at work](https://books.google.com/books/about/Team_Roles_at_Work.html?id=hF2yJzYfUBAC) and link to AI for each role.

Cons: \
Misleading link proofs for 6 out of 9 Belbin Team Roles

**Summary:**

AI to Improve organization performance based on nine Belbin Team roles, however relevant proofs not provided for 6 out of 9 roles on this theoretical paper

**Comments And Feedback To The Authors:**

Paper requires major revision to include precise proof & proper conclusion to be CCR
1. This paper is a good start considering one of the author matches the URM criteria.
2. Considering this is a theoretical paper, relevant proofs for the assumptions and concise conclusion is of utmost importance.
3. On complete look into the links provided as assumptions proofs in the 9 Belbin Team roles, only 3 out of 6 roles is relevant.
4. Kindly recheck and provide actual proofs and make necessary changes for the roles: _Shaper_, _Implementer_, _Specialist_, _Completer-Finisher_, _Resource-Investigators_
5. _Coordinator_- mentions this [link](https://www.besci.org/tactics/gamification). Kindly add a line with example of software that does this.

**Reason For Not Giving A Higher Recommendation:**

1. Though the paper mentions a __novel approach__ introduced, it fails to mention any novel approach.
2. Assumptions are made based on 9 Belbin Team Roles without any relevant proof for many of the roles.
Only 3 out of 9 roles are provided with actual proof: _Plant_, _Monitor-Evaluator_, _Team Worker_

   * __Shaper__- executive who sets vision & manage crisis proof reference to  ML System Safety in an environment: robustness, monitoring, alignment, and systemic safety. The proof link does not manage crisis or set visions
   * __Implementer__- no proof provided
   * __Specialist__- examples provided are for content creation(blogs in social media) & analysing syntactic constructs of Java programs.. The examples are not relevant for the role of a Specialist who has domain-specific technical skills like programming
   * __Completer-Finisher__: Grammarly proof is not provided, but instead proof link points to of Quantitative risk assessment of typhoon storm surge for multi-risk sources
   * __Resource-Investigators__ who build partnership, outreach & new connections
     * recommender system for potential partner proof points to documentation on planning AI marketing strategy( CMOs classify existing AI projects and plan the rollout of future ones). This is not a recommender proof but of planning a project
     * Proof of Conversations points to AI calendar which assists in estimating time of task completion

**Reason For Not Giving A Lower Recommendation:**

N/A

---

### Decision · Program_Chairs · 2023-04-08

No revision received; not invited to archive